# Endocrine Disruptor Bisphenol A (BPA) Triggers Systemic Para-Inflammation and is Sufficient to Induce Airway Allergic Sensitization in Mice

**DOI:** 10.3390/nu12020343

**Published:** 2020-01-28

**Authors:** Lucas Fedele Loffredo, Mackenzie Elyse Coden, Sergejs Berdnikovs

**Affiliations:** Division of Allergy and Immunology, Department of Medicine, Northwestern University Feinberg School of Medicine, Chicago, IL 60611, USA; lucasloffredo@gmail.com (L.F.L.); mackenzie.coden@northwestern.edu (M.E.C.)

**Keywords:** bisphenol A, estrogen, xenoestrogens, para-inflammation, endocrine, alarmins, allergy, asthma

## Abstract

Allergic airway diseases are accompanied by increased permeability and an inflammatory state of epithelial barriers, which are thought to be susceptible to allergen sensitization. Although exogenous drivers (proteases, allergens) of epithelial barrier disruption and sensitization are well studied, endogenous contributors (diet, xenobiotics, hormones, and metabolism) to allergic sensitization are much less understood. Xenoestrogens are synthetic or natural chemical compounds that have the ability to mimic estrogen and are ubiquitous in the food and water supply of developed countries. By interfering with the estrogen produced by the endocrine system, these compounds have the systemic potential to disrupt the homeostasis of multiple tissues. Our study examined the potential of prototypical xenoestrogen bisphenol A (BPA) to disrupt epithelial homeostasis in vitro and promote allergic responses in vivo. We found that BPA exposure in epithelial cultures in vitro significantly inhibited epithelial cell proliferation and wound healing, as well as promoted the expression of the innate alarmin cytokine TSLP in a time-and dose-dependent manner. In vivo, the exposure to BPA through water supply or inhalation induced a systemic para-inflammatory response by promoting the expression of innate inflammatory mediators in the skin, gut, and airway. In a murine tolerogenic antigen challenge model, chronic systemic exposure to BPA was sufficient to induce airway sensitization to innocuous chicken egg ovalbumin in the complete absence of adjuvants. Mechanistic studies are needed to test conclusively whether endocrine disruptors may play an upstream role in allergic sensitization via their ability to promote a para-inflammatory state.

## 1. Introduction

The epithelium is the first barrier encountered by an inhaled or ingested allergen [1]. Allergic inflammatory diseases are accompanied by the increased permeability and inflammatory state of the epithelial barrier, which is thought to be more susceptible to allergen sensitization [2,3,4,5]. Multiple lines of evidence point to the causality of epithelial barrier dysfunction in the development of allergic inflammation. A number of mouse and human models using protease epithelial damage triggers or the targeted deletion of structural or junctional barrier genes report enhanced allergic sensitization and Th2 inflammation [6,7,8,9]. Although research shows that exogenous proteases in many allergens themselves are sufficient to disrupt epithelium, this does not explain why only a fraction of the population exposed to the same allergens develops sensitization. Endogenous factors linked to systemic epithelial barrier dysfunction, such as changes in nutrients, hormones, vitamins, chemical exposures, and dysbiosis, are suspected in the origins of allergic disease but are much less understood [10]. Among them, xenoestrogens are receiving attention due to their biological action and ubiquitous human chronic exposure through their release into beverages, food, and the environment [11,12,13]. Xenoestrogens are synthetic or natural chemical compounds that have the ability to mimic estrogen and have estrogenic effects on biological organisms, thus interfering with the estrogen produced by the endocrine system. For this reason, they are also sometimes called “environmental hormones” or “endocrine disrupting compounds”. Synthetic xenoestrogens are widely used industrial compounds that are prevalent in the food and water supply of developed countries because of their widespread use as plasticizers in the production of food packaging. Low but persistent non-toxic exposure to xenoestrogens has been recently shown to be a serious environmental hazard linked to a vast array of human conditions, including allergic diseases [14,15,16,17]. Estrogen receptors alpha and beta play a central role in epithelial homeostasis in both sexes, which is independent of their role in reproduction and sexual development [18,19,20]. In particular, estrogen receptors are essential in the integration of extracellular signals, such as growth factor and WNT/Notch pathways, to properly regulate the expression of genes that control cell fates during epithelial turnover [20,21,22,23,24,25]. In this manuscript, we tested whether the ubiquitous endocrine disruptor bisphenol A (BPA) is disruptive for epithelial homeostasis with possible in vivo consequences for the initiation of allergic responses. We found that BPA exposure (1) significantly inhibited epithelial cell proliferation and wound healing in vitro as well as promoted the expression of innate alarmin cytokine TSLP in a time- and dose-dependent manner; (2) elevated the systemic expression of innate cytokines and chemokines at skin, gut, and airway barriers when given to mice in water or by inhalation; and (3) in tolerogenic airway antigen challenge protocol was sufficient to induce sensitization to chicken egg ovalbumin in the absence of adjuvants.

## 2. Materials and Methods 

### 2.1. Animal Experiments 

For all in vivo asthma model experiments, we used wild-type adult female littermate mice on the BALB/cJ background (Jackson Laboratories, Bar Harbor, ME, USA). The Institutional Animal Care and Use Committee of Northwestern University approved all animal procedures (protocol IS00001710, original approval date 6/8/2015). For the in vivo experiments, bisphenol A (BPA) (Sigma) was administered at 25 mg/L in drinking water bottles *ad libitum* or intranasally (i.n.) daily at 25 ug/mL (volume administered = 53 µL). This BPA concentration is comparable to human environmental exposure of 5 mg/kg of body weight/day, which is the current NOAEL (no observed adverse effect level) concentration set by the U.S. Food and Drug Administration (FDA) and the European Food Safety Authority (EFSA). This is also the dose typically used in murine studies of low-dose BPA exposures. The BPA was not specifically endotoxin free, but it was ≥ 99.0 pure by HPLC (Sigma). Lung, skin (abdominal), and gut (duodenal) tissues were harvested after seven days of BPA exposure. For the allergic model, mice were maintained on regular drinking water or water with 25 mg/L BPA for 70 days to mimic chronic exposures by ingestion. In the last 10 days of the protocol, mice were administered daily saline or 1% chicken egg ovalbumin (OVA) grade VI (Sigma) with or without BPA (25 ug/mL) by intranasal inhalation. We obtained bronchoalveolar lavage fluid (BALF), lung tissue, and serum 24 h after the last inhalation challenge.

### 2.2. Bronchoalveolar Lavage, Lung Digestion, and Flow Cytometry

Bronchoalveolar lavage was performed by lavaging lungs with ice-cold 1× phosphate-buffered saline (PBS) through the cannulated trachea. Lungs were digested in 1 mg/mL collagenase D and 0.2 mg/mL DNAse I (Roche, Indianapolis, IN, USA) in preparation for flow cytometry. Digested tissue was filtered through sterile mesh and incubated in 1× BD PharmLyse Lysing Buffer (BD Biosciences, San Jose, CA, USA) to lyse red blood cells. Live/dead exclusion was performed using Aqua dye (Molecular Probes) followed by incubation with CD16/CD32 FC Block (BD Pharmingen, San Jose, CA, USA). Antibody cocktail was added directly to blocked samples and incubated for 30 min at 4 °C. Antibody cocktail composition was as previously used for leukocyte population characterization during allergic inflammation [26]. Samples were acquired on a BD LSRII flow cytometer (BD Biosciences). BAL cells were pelleted by centrifugation at 300× g for 5 min, washed, and prepared as described above, starting with the live/dead step. Bead compensation (OneComp; eBioscience, San Diego, CA, USA, and ArC; Molecular Probes beads), gating, and data analysis were performed using FlowJo v.10 (TreeStar, Inc., Ashland, OR, USA). Only live, single, hematopoietic (CD45+) cells were used in all analyses. Fluorescence Minus One (FMO) controls were used to set up gate boundaries. Leukocyte populations were identified as follows: (i) eosinophils: CD11b(+)Ly6G(low/-)CD11c(−/low)Siglec-F(med/high); (ii) alveolar macrophages: CD11b(−)Ly6G(−)CD11c(high)Siglec-F(high); and (iii) neutrophils: CD11b(+)Ly6G(high)CD11c(−)Siglec-F(−).

### 2.3. Cell Culture

For epithelial cultures, we used the commercially available (Sigma) BEAS-2B human cell line originally derived from normal bronchial epithelium obtained from the autopsy of non-cancerous individuals. Cells were cultured in Dulbecco’s Modified Eagle Medium (DMEM) with the addition of 1% penicillin–streptomycin and 5% heat-inactivated fetal bovine serum. Cells were treated with BPA dissolved in ethanol (ethanol alone used for vehicle controls). Cells were passaged when reaching 70%–80% confluency to avoid spontaneous transformation. For wound-healing assays, epithelial monolayers were scratched using p10 pipet tips, and wound closure was monitored by bright field microscopy using same field view over a 48 h period. Scratch width was quantified using ImageJ software (NIH). 

### 2.4. Quantitative PCR

RNA was isolated from cells using the Qiagen RNeasy mini kit (Qiagen). cDNA was synthesized using a qScript cDNA synthesis kit (Quanta BioSciences) and analyzed by real-time PCR on a 7500 real-time PCR system (Applied Biosystems) using primers/probes from Integrated DNA Technologies and PrimeTime Gene Expression Master Mix (IDT or Applied Biosystems).

### 2.5. Cytotoxicity Assay

The Vybrant Cytotoxicity Assay Kit (Thermo Fisher), which detects glucose-6-phosphate dehydrogenase released from damaged cells via the reduction of resazurin into red-fluorescent resorufin, was used to assess cytotoxicity. A total of 7000 BEAS-2B cells were plated in 50 uL of normal media in a 96-well plate and incubated for 24 h to adhere; then, they were changed to 50 uL media with indicated BPA concentrations and incubated for another 24 h. Control wells were lysed with cell lysis buffer from the kit; then, 50 uL of resazurin/enzymatic solution was added to each well, and each plate was incubated on a shaker in the dark for 10 min. Then, fluorescence was measured on a fluorescent plate reader (excitation 560 nm, emission 600 nm).

### 2.6. Measurements of Serum Proteins and Antibodies

Harvested serum was assayed for cytokine IL-33 using a Ready-SET-Go! ELISA kit purchased from eBioscience (Invitrogen). ELISA assays were performed according to the manufacturer’s instructions. OVA-specific IgE was determined by custom ELISA, as previously described [27].

### 2.7. Statistical Analysis

Statistical significance of all data was determined by an unpaired t-test or one-way ANOVA followed by Tukey’s post hoc pairwise testing whenever applicable. All data are represented as mean ± S.E.M. Statistical analysis was performed using GraphPad Prism 7 (GraphPad Software, Inc.). An alpha level of 0.05 was used as a significance cut-off in all tests. For principal component analysis (PCA), PAST v.3 software was used, inputting data as log10-normalized values from the qPCR of alarmin, cytokine, and chemokine gene expression (genes used: *Il33, Tslp, Ifng, Tnfa, Il10, Cxcl9, Cxcl10, Cxcl11, Ccl1, Ccl2,* and *Ccl11*) from the following nine sample groups: non-treated lung tissue, lung tissue from BPA water-treated mice, BPA-i.n. treated lung tissue, non-treated gut tissue, BPA-water treated gut tissue, BPA-i.n. treated gut tissue, non-treated skin tissue, BPA-water treated skin tissue, and BPA-i.n. treated skin tissue.

## 3. Results 

### 3.1. Bisphenol A Has An Inhibitory Effect on BEAS-2B Epithelial Cell Proliferation and Wound Healing

Estrogen receptors intricately interact with developmental pathway signaling to maintain epithelial homeostasis [21]. Using wound scratch assays, we found that BPA had an inhibitory effect on epithelial wound healing (Figure 1A,B). Our data show that this effect of BPA was likely mediated by the significant inhibition of epithelial cell proliferation, which was both time- and concentration-dependent (Figure 1C). The disruption of epithelial growth by BPA occurred without detectable cytotoxicity at concentrations less than 200 µM (Figure 1E). It is likely that BPA could disrupt epithelium in a non-damaging manner via interfering with the estrogen regulation of epithelial junctions and cell cycle. Moreover, we detected a significant upregulation of the *Tslp* message by BPA-treated epithelial cells in vitro, which was especially evident after 48 h of exposure in culture (Figure 1D). Notably, the highest TSLP expression at 200 µM was at least partially associated with BPA cytotoxicity, which thus can serve as a positive control for levels of death-induced alarmin expression. In summary, BPA directly interferes with homeostatic proliferation and promotes TSLP expression by epithelial cells.

### 3.2. Ingestion of BPA Promotes Systemic Para-Inflammation in Mice

Given the observed effects on epithelial cells in vitro, we proceeded to confirm this in vivo by exposing mice to BPA (5 mg/kg of body weight/day) in water *ad libitum* for seven days (Figure 2A). Another group of mice were exposed to BPA via intranasal inhalation daily at a concentration of 25 ug/mL. The local exposure of epithelial barriers to BPA (airway by inhalation and intestinal barrier by ingestion) promoted the expression of *Ccl1* and *Ifnγ* at several contact sites (Figure 2B). Surprisingly, this was accompanied by significant changes in the expression of these mediators and trends for changes in the expression of multiple other mediators at other barrier sites as well, which were not the result of direct exposure to BPA (Figure 2B,C). In Figure 2C, we used exploratory principal component analysis (PCA) to summarize the variation in expression of all genes measured by qPCR (regardless of significance) at three barrier sites following BPA exposures via water supply or inhalation. It suggests that each barrier site (lung, gut, skin) may promote the expression of innate cytokines and chemokines even if not exposed to BPA directly (Figure 2C), which is consistent with low-grade systemic inflammation.

### 3.3. Chronic Systemic BPA Exposure Induces Allergic Sensitization to Innocuous OVA Antigen Exposure and Facilitates the Development of Allergic Inflammation

We further tested whether the para-inflammatory state induced by BPA exposure would facilitate sensitization to chicken egg ovalbumin (OVA) in a tolerogenic antigen exposure protocol. The treatment of mice with 25 ug/mL of BPA in water supply for 10 weeks followed by the daily intranasal treatment of endotoxin-free 1% OVA for 10 days resulted in spontaneous, adjuvant-free sensitization and allergic inflammation development (Figure 3). This OVA treatment protocol is completely tolerogenic and does not result in sensitization or allergic inflammation in the absence of adjuvants. Although there was some non-specific inflammatory response and neutrophil recruitment in control mice (receiving water only, no BPA) challenged with 1% OVA, eosinophils were significantly recruited only in mice treated with BPA plus OVA (Figure 3). Interestingly, the eosinophils in this treatment group showed a CD11c (+) phenotype indicative of their mucosal activation and heightened capacity to migrate to the airway [26] (Figure 3B). The finding that BPA was sufficient to elicit an allergic inflammatory response to innocuous OVA antigen in the complete absence of adjuvant was reinforced by the demonstration that the lung expression of Type 2 cytokines interleukin (IL)-4, IL-5, and IL-13 was significantly upregulated only in mice administered BPA and OVA (Figure 4A). Consistent with the inflammatory response and eosinophil recruitment, we also observed the lung tissue expression of chemokines CCL2 (MCP1) and CCL11/CCL24 (eotaxins 1 and 2) (Figure 4B). There was no difference in the expression of cytokine IL-33 in the lung tissue of mice challenged with BPA and OVA; however, we found significantly elevated serum protein levels of IL-33 in these mice (Figure 4C). Interestingly, BPA administration alone (without OVA) resulted in the marginally significant elevation of IL-33 protein serum, further demonstrating its systemic potential (Figure 4C). To fully demonstrate that BPA promoted allergic sensitization, we measured OVA-specific IgE antibodies in serum. Again, only mice receiving both BPA and OVA showed significant antigen sensitization (Figure 4D).

## 4. Discussion

In this study, we tested whether xenoestrogen BPA, which is ubiquitous in the food and water supply of developed countries, promotes the endogenous systemic disruption of epithelial barriers and contributes to the initiation of allergic responses. With a total worldwide production capacity exceeding 6 billion pounds per year, BPA is one of the highest volume chemicals in commercial production today due to its widespread use in the production of plastics and flame retardants [14]. The ester bond linking BPA molecules in polycarbonates and resins undergoes hydrolysis, resulting in the release of low levels of free BPA into food, the water supply, and the environment. Measurements of unconjugated BPA in human blood, tissues, and urine in the United States, European Union, and Japan show higher than predicted levels, suggesting continuous exposure to significant amounts of BPA [14]. Normal functioning of the endocrine system is critical in the maintenance of multiple systemic homeostatic processes, including metabolism, normal neuroendocrine and immune function, and tissue homeostatic maintenance and renewal [28,29,30]. Sex steroids, in particular estrogens (aside from their function in reproduction and sexual development), are critical in maintaining systemic homeostasis in both sexes [19,28,31,32,33]. Given the centrality of hormones in the maintenance of systemic homeostasis, it is not surprising that epidemiological and experimental research studies link endocrine disruption by xenoestrogens to inflammatory diseases, cancer, metabolic syndrome, and neuroendocrine and reproductive abnormalities [34,35,36]. Mouse models suggest that BPA exacerbates allergic inflammation [37,38,39,40] and that maternal exposure to BPA enhances the development of allergic inflammation in offspring [15]. There are multiple epidemiological studies linking bisphenol exposures (typically measured in urine) to the development of asthma and other allergic conditions [41,42,43]. For example, higher postnatal urinary BPA concentrations were associated significantly with asthma in inner-city children [16]. The concentrations of BPA that we used in this study likely correspond only to high-end human exposures, since an acceptable daily human intake of BPA is typically 1000-fold below the NOAEL, and human serum levels of BPA range between 0.2 and 20 ng/mL [44]. However, our study was not aiming to represent daily human intake, but rather to add to the discovery of immune processes driven by “low-dose” BPA exposures typical for murine studies.

Tissue and epithelial homeostasis is maintained by balancing cell proliferation, differentiation, and death. It is well established that the nuclear receptor superfamily controls homeostasis by mediating the regulatory activities of many hormones, growth factors, and metabolites [45,46,47,48]. Estrogen receptors were shown to intricately interact with the WNT and Notch developmental pathways for the maintenance of epithelial homeostasis [21]. Consistent with this, we found that BPA could disrupt epithelium in a non-damaging manner via interfering with proliferation. Moreover, epithelial cells cultured with BPA expressed innate alarmin cytokine TSLP, which was also observed in study of BPA epithelial biology by Tajiki-Nishino et al. [37]. In support of these observations, it has been published that Notch-deficient keratinocytes fail to differentiate and release high levels of TSLP, which is critical to the development of allergic diseases [49]. More studies in the literature suggest that chronic BPA exposure could also affect the proliferation of epithelial basal progenitor cells [50]. Epithelial cells are known to produce innate cytokines in response to numerous stimuli [51]. The knockdown of filaggrin and E-cadherin induces the mRNA expression of TSLP through the epidermal growth factor receptor (EGFR) signaling pathways [52,53]. Importantly, Notch signaling and cellular receptors (protease-activated receptors (PARs), retinoic acid, and peroxisome proliferator-activated receptors (PPARs)) have regulatory activity for TSLP [49,51]. Studies of estrogen show similar effects to the results reported here for BPA, which is likely because both signal via a common pathway. In particular, estrogen has been demonstrated to induce the secretion of TSLP from human endometrial stromal cells in a dose-dependent manner [54]. Estrogen has also been shown to negatively regulate epithelial wound healing in multiple mouse and human studies [55,56,57,58]. Whether steroid hormones and xenoestrogens directly regulate TSLP and alarmin production by human epithelium warrants further investigation.

While evaluating in vivo epithelial barrier responses (skin, gut, airway) to systemic BPA exposure, we found that BPA significantly promoted the expression of Ccl1 and Ifnγ at more than one barrier site. Multiple mediators that are regulatory for the tissue innate immune system showed trending but not significant expression consistent with low-grade inflammatory response. Among them, chemokine Ccl1 is known to promote the recruitment of monocytes and eosinophils to tissue during the development of allergic inflammation [59]. Cxcl9, Cxcl10, and Cxcl11 are a family of interferon gamma-induced chemoattractant proteins stimulatory for the recruitment of monocytes, dendritic cells, and T cells to tissue. The induction of such responses is known to facilitate the inception of Type 2 sensitization [60]. Moreover, we found a significant increase in serum IL-33 protein levels after BPA exposure only. Such systemic action of BPA reported by us and others is consistent with the concept of a para-inflammatory response [61], which is likely mediated by its systemic interference with estrogen homeostatic signaling. Para-inflammation is a tissue adaptive response to persistent stress (distinct from direct injury and infection) to restore tissue functionality and homeostasis. It is thought to underlie the chronic inflammatory conditions associated with modern human diseases [62]. 

In our mouse experiments, such a systemically induced para-inflammatory state was sufficient to promote spontaneous, adjuvant-free sensitization as well as development of allergic inflammation. Although several previous studies examined the effect of oral and/or intratracheal BPA exposure on allergic responses in juvenile and adult mice, BPA was reported only as an exacerbating or aggravating factor in the development of allergic airway inflammation [37,38,39,40]. However, they parallel our results by showing the inflammatory-inducing potential of BPA in vivo via different exposure routes and the lack of necessity for standard adjuvants in these models. Our study, using a tolerogenic exposure protocol to low doses of highly purified OVA antigen, confirms that the ingestion of BPA is not only aggravating but is a sufficient factor to facilitate allergic sensitization. Thus, we would like to bring to attention that BPA may play an upstream role in sensitization via its ability to promote a para-inflammatory state, which warrants further mechanistic investigation. It is likely that multiple xenoestrogens, as well as other natural and systemic chemicals in our food supply, are capable of inducing similar systemic effects, possibly at different concentration ranges or exposure durations. Our study used bisphenol A as a prototypical xenoestrogen to emphasize the systemic inflammatory potential of endogenous endocrine dysregulation and suggest its potentially critical upstream role in promoting allergic responses.

## 5. Conclusions

The Endocrine disruption potential of BPA stems from its potential to interfere with epithelial homeostatic signals regulated by estrogen. We show that BPA exposure interferes with epithelial proliferation and triggers innate inflammatory responses from epithelial cells in vitro and systemically in vivo. Such systemic para-inflammatory response would present fertile ground for allergic sensitization, which we indeed observed in our brief study. Further mechanistic studies are needed to test conclusively whether endocrine disruption may play an upstream role in allergic sensitization via induction of a para-inflammatory state.

## Figures and Tables

**Figure 1 nutrients-12-00343-f001:**
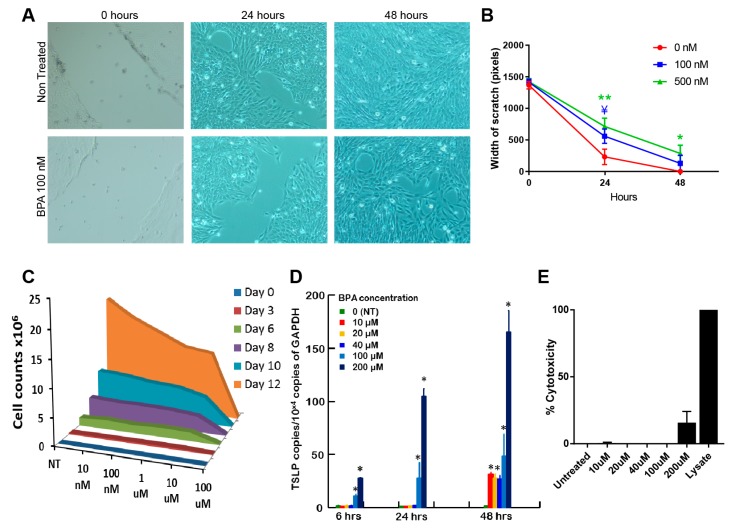
Bisphenol A has an inhibitory effect on BEAS-2B epithelial cell proliferation and wound healing. (**A**) Wound healing assay. Cells were grown to complete confluency, and monolayers were scratched with a p10 pipet tip. Wound closure was monitored over a 48 h period. Representative images from two experiments. (**B**) Quantification of wound closure rates using scratch width measured in ImageJ. Representative quantification from two experiments. (**C**) Inhibitory effect of bisphenol A (BPA) on long-term BEAS-2B epithelial cells proliferation in culture. Data shown from experiment performed one time. (**D**) BEAS-2B expression of TSLP induced by exposure to BPA. Representative quantification from experiment performed three times. (**E**) Cytotoxicity analysis of BEAS-2B epithelial cell exposure to BPA; cells were cultured for 24 h for adherence, treated with indicated concentrations of BPA for 24 h, and then assessed for cytotoxicity. All values included from experiment performed two times. *, *p* < 0.05 by ANOVA.

**Figure 2 nutrients-12-00343-f002:**
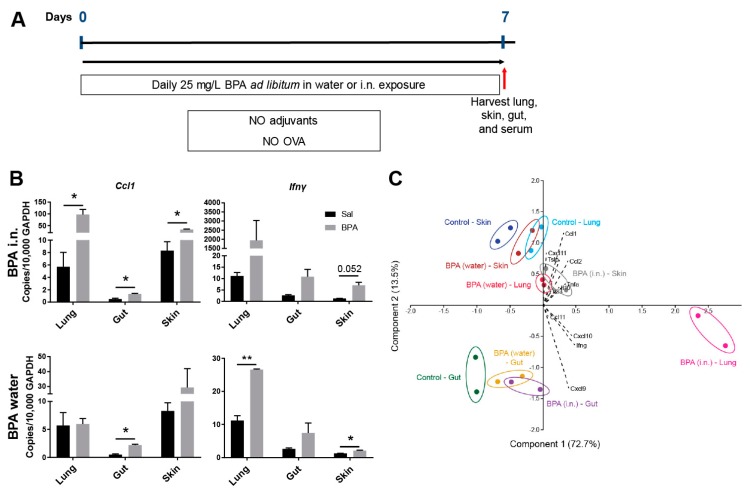
Ingestion of BPA promotes systemic para-inflammation in mice. (**A**) BPA administration protocol. **(B**) Expression of *Ccl1* and *Ifng* in murine lung, gut, and airway after seven days of BPA exposure. Top graphs, exposure by intranasal inhalation; bottom graphs, exposure through *ad libitum* water intake. N = 2 mice/group/9 groups, all data are from one experiment. *, *p* < 0.05, **, *p* < 0.01 by t-test within each tissue compartment. (**C**) Exploratory PCA analysis of log10-normalized values of gene expression measured at all tissue sites (lung, skin, gut) following BPA exposure via water intake or by inhalation (see Methods for list of genes/groups).

**Figure 3 nutrients-12-00343-f003:**
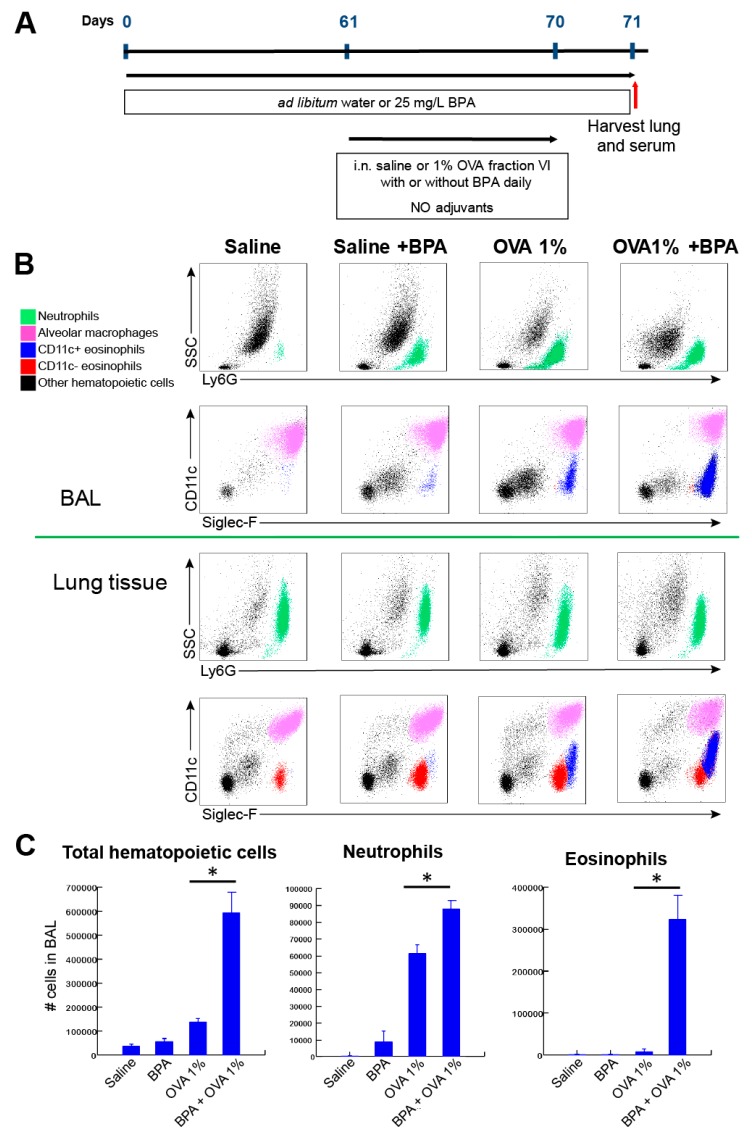
Chronic systemic exposure to BPA facilitates the development of allergic inflammation in a murine tolerogenic ovalbumin (OVA) treatment protocol. (**A**) BPA administration and OVA antigen challenge treatment timeline. (**B**) Flow cytometry analysis of leukocyte inflammatory response. Top charts, leukocyte populations measured in bronchoalveolar lavage; bottom charts, leukocyte responses measured in homogenized lung tissue. (**C**) Quantification of numbers of recruited cells in BALs by flow cytometry. N = 2 in saline group, N = 3-4 mice in treatment groups, all data shown from experiment performed one time. *, *p* < 0.05 by ANOVA.

**Figure 4 nutrients-12-00343-f004:**
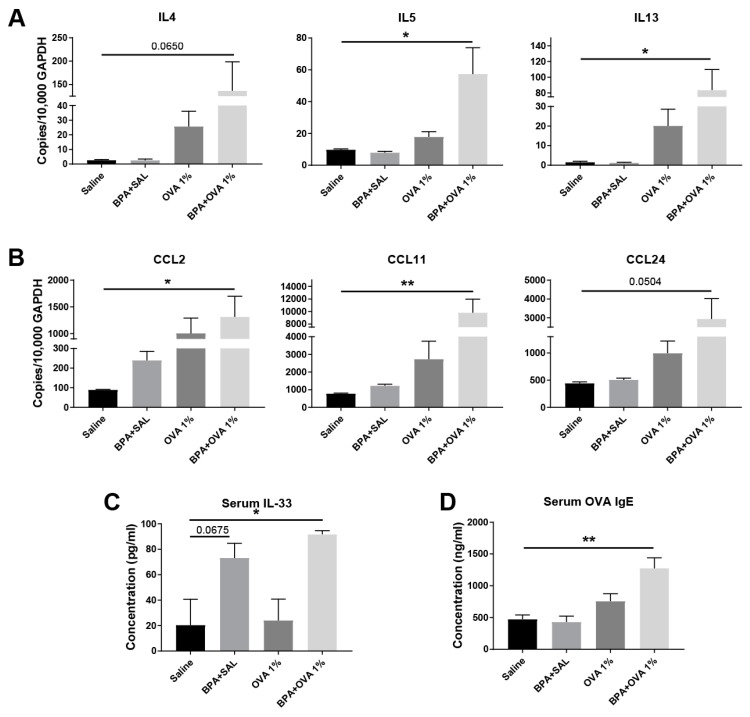
Chronic systemic BPA exposure induces allergic sensitization to innocuous OVA antigen exposure followed by the expression of Type 2 immune mediators. (**A**) Lung tissue expression of Type 2 cytokines by qPCR. (**B**) Lung tissue expression of chemokines by qPCR. (**C**) Serum protein levels of alarmin cytokine IL-33 by ELISA. (**D**) Serum levels of OVA-specific IgE antibodies by ELISA. N = 2 in saline group, N = 3-4 mice in treatment groups; all data shown from an experiment performed one time. *, *p* < 0.05, **, *p* < 0.01 by ANOVA.

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
