# Peer review of "Endocrine Disruptor Bisphenol A (BPA) Triggers Systemic Para-Inflammation and is Sufficient to Induce Airway Allergic Sensitization in Mice"

_nutrients, 2020, doi:10.3390/nu12020343_

Round 1

Reviewer 1 Report

This is a well written and well performed study on effects of BPA on the triggering of systemic para-inflammation and induction of airway allergenic sensitization in mice.

Information on intranasal exposure regime is lacking from the material and method. Was it one time exposure through instillation or given in a more chronic regime? Please explain the intranasal exposure regime. Information on the purity of BPA is lacking. Is it endotoxin-free? How would a positive control/known xenoestrogen/estrogen exposure be compared to BPA regarding BEAS-2B epithelial cell proliferation, wound healing and TSLP expression? Order of the figures, starting with figure 2, 3, and thereafter number 1. Please correct BPA exposure in the cell culture, how is this exposure/concentration chosen in relation to human relevant exposure? How about CXCL12 expression, was that analyzed? From where were the human BEAS-2B cell line obtained?

Reviewer 2 Report

This is an interesting, well-written paper about a relevant topic. I only have some small remarks.

line 73: Hazard assessments by major regulatory and advisory bodies are in agreement that the overall no-observed-adverse-effect level (NOAEL) for BPA from robust data is 5 mg/kg body weight/day. This is minimally five hundred-fold above conservative estimates of human exposure, including in bottle-fed infants. I think the exposure level should also be addressed. line 75: add how the intranasal exposure was performed (with or without anaesthesia, route, volume and concentration BPA). line 130: Cytotoxicity is very important for the interpretation of these results. They should be available within in the paper or in a supplementary file. Fig 2B. IFNg after i.n. exposure seems to be statistically different on eye, however this is not indicated. Is this correct? Why is it then depicted in fig 2C? Fig 2 C: the quantified data should be made available. Were these cytokines significantly changed? Did you also measure TSLP in lung tissue after OVA/BPA exposure? Was this consistent with the in-vitro data or could no differences be measured in total lung tissue? This could be explained by the “contamination” of the epithelial cells with other cell types. Are there any epidemiological studies confirming or refuting the experimental results presented here?
